# Evaluation of Adjunctive Aminoglycoside Therapy Compared to β-Lactam Monotherapy in Critically Ill Patients with Gram-Negative Bloodstream Infections

**DOI:** 10.3390/antibiotics14050497

**Published:** 2025-05-13

**Authors:** Joshua Eudy, Aaron M. Chase, Divisha Sharma, Zoheb Irshad Sulaiman, August Anderson, Ashley Huggett, Lucy Gloe, Daniel T. Anderson

**Affiliations:** 1Department of Pharmacy, Wellstar MCG Health, Augusta, GA 30912, USA; 2Department of Clinical and Administrative Pharmacy, University of Georgia College of Pharmacy, Athens, GA 30602, USA; 3Department of Medicine, Division of Infectious Diseases, Medical College of Georgia, Augusta University, Augusta, GA 30912, USAdanderson3@augusta.edu (D.T.A.); 4Medical College of Georgia, Augusta University, Augusta, GA 30912, USA; 5Department of Pharmacology & Toxicology, Medical College of Georgia, Augusta University, Augusta, GA 30912, USA

**Keywords:** bloodstream infection, intensive care, aminoglycoside, antimicrobial stewardship

## Abstract

**Background/Objectives**: Gram-negative bloodstream infections (GN-BSIs) in the critically ill carry significant mortality, which is exacerbated by delays in appropriate therapy. To improve the time to effective therapy, aminoglycosides are often recommended as empiric adjunctive antimicrobials. However, there is a paucity of clinical data supporting this practice. This study’s objective was to evaluate the safety and efficacy of adjunctive aminoglycosides compared to β-lactam monotherapy in patients admitted to the intensive care unit (ICU) with GN-BSI. **Methods**: This was a retrospective, propensity-matched cohort study of critically ill patients with GN-BSI. The primary outcome was 15-day all-cause mortality. The secondary endpoints evaluated included 30-day mortality, ICU-free survival days, 60-day relapse, 30-day readmission, development of acute kidney injury (AKI), and new resistance. **Results**: A total of 209 propensity-matched patients were included for analysis: 136 received β-lactam monotherapy and 73 received adjunctive aminoglycoside. The primary outcome of 15-day all-cause mortality was not significantly different between groups (17% vs. 21%; *p* = 0.644). Additional secondary endpoints of 30-day mortality (22% vs. 25%), ICU-free survival (12.1 vs. 12.2 days), 60-day relapse (3.3% vs. 7.4%), and 30-day readmission (23% vs. 18%) did not yield significant differences. The proportion of AKI was higher in the adjunctive aminoglycoside group but was not found to be significantly different (26.5% vs. 37%). **Conclusions**: The use of adjunctive aminoglycosides for GN-BSI did not affect clinical outcomes in the critically ill.

## 1. Introduction

Gram-negative bloodstream infections (GN-BSIs) represent a significant burden on public health due to their increasing prevalence and high mortality [1,2]. Further complicating the management of these infections is the rapid development of antimicrobial resistance among Gram-negative organisms [3]. For these reasons, empiric combination therapy for Gram-negative infections in high-risk populations, such as critically ill patients, is often recommended by medical societies and affiliated consensus guidelines, such as the International Surviving Sepsis Campaign and the Infectious Diseases Society of America (IDSA) [4,5]. However, recommendations for combination therapy are stated as weak and based on low-quality evidence. The lack of supporting evidence and conflicting data puts clinicians in a difficult position of trying to balance the importance of providing adequate coverage in high-risk patients while attenuating the risks of adverse effects.

Combination therapy, targeting Gram-negative organisms, is often defined as the use of a broad-spectrum beta-lactam in addition to an aminoglycoside or a fluoroquinolone active against *Pseudomonas aeruginosa*. Due to increasing rates of resistance to fluoroquinolones, aminoglycosides are often used as a first-line therapy. Historically, there have been three proposed advantages to using aminoglycoside combination therapy: (i) potential synergy with β-lactams, (ii) prevention of future development of multidrug-resistant organisms (MDROs), and (iii) broadened Gram-negative coverage leading to an increased likelihood of at least one antimicrobial being active against suspected or confirmed pathogens [6]. Regarding synergy, there are historical data demonstrating benefit in vitro, but there does not appear to be a significant benefit in clinical outcomes [7,8,9,10,11,12,13,14,15,16]. Further, the effect of synergistic relationships does not appear to be as beneficial as once believed. Newer data suggest that depending on the combination of β-lactam or aminoglycoside used for a specific organism, there may be potential antagonistic effects [17]. Secondly, clinical studies do not provide evidence that this practice reduces the emergence of antimicrobial resistance [18,19]. This leaves the increased likelihood of having at least one active antimicrobial with in vitro activity as a common rationale for combination therapy in recent years [20,21,22]. However, this rationale is not well supported, as evidenced by the conflicting clinical data on the effects of combination therapy [23], even in high-risk populations [24,25,26]. In part, this could stem from pharmacokinetic/pharmacodynamic [PK/PD] issues, which have led to recent breakpoint changes from the Clinical Laboratory and Science Institutes [CLSI], resulting in a significant increase in resistance to aminoglycosides.

The conflicting data on clinical outcomes, possible antagonistic effects, and risk of nephrotoxicity warrant further investigation into the utility of adjunctive aminoglycoside therapy. This study aims to evaluate empiric β-lactam monotherapy versus adjunctive aminoglycoside therapy in one of the highest-risk populations: critically ill patients with Gram-negative bloodstream infections.

## 2. Results

### 2.1. Demographics

A total of 209 propensity-matched patients—136 who received β-lactam monotherapy and 73 who received adjunctive aminoglycoside—were included from a cohort of 439 unmatched patients (Table 1). Analysis of the propensity-matched cohort baseline characteristics demonstrated good categorical agreement [standard mean difference (SMD) ≤ 0.1] (Table 1). In both groups, the urinary tract was the predominant source of infection, and *Escherichia coli* was the most frequently isolated organism. A Pitt bacteremia score > 4, Charlson comorbidity index (CCI) (5.7 vs. 5.9), and vasopressor use in 81% of patients were all representative of a critically ill population and were similar across the matched cohorts. Notable exceptions between the β-lactam monotherapy and adjunctive aminoglycoside groups included incidence of MDR organisms (21% vs. 26%; SMD 0.111), intra-abdominal source (14% vs. 22%; SMD 0.274), primary BSI source (19% vs. 10%; SMD 0.274), and source control (44% vs. 71%; SMD 0.588).

Regarding aminoglycoside therapy use, gentamicin was predominantly used (91%) and was continued on average for 1.5 doses. With institutional protocols prioritizing extended interval dosing, this would correspond to approximately two days of drug therapy. However, this duration of therapy could be longer based on renal insufficiency. The mean aminoglycoside dose was 4.2 mg/kg (SD 1.6 mg/kg). The median time from index culture to first dose was 18 h (Table 2).

### 2.2. Outcomes

The 15-day all-cause mortality was similar between patients who did not receive combination aminoglycosides and those who did (17% vs. 21%; *p* = 0.644). Further, there was no difference in 30-day all-cause mortality (22% vs. 25%), ICU-free survival (12 days vs. 12 days), 60-day relapse (3% vs. 7%), or 30-day readmission (23% vs. 19%) between the matched cohorts. The number of patients that developed new resistance was low in both cohorts. The development of AKI was higher in the aminoglycoside cohort but not significantly different from those who did not receive aminoglycosides (27% vs. 37%; *p* = 0.155) (Table 3).

Kaplan–Meier curves evaluating mortality showed no significant differences between either the unmatched (Figure 1) or matched cohorts (Figure 2). Similarly, in a subgroup analysis of MDROs, no significant differences were observed between those who received adjunctive aminoglycoside compared to those who received β-lactam monotherapy (Figure 3).

## 3. Discussion

In this study of propensity-matched, critically ill patients with GN-BSI, the use of adjunctive aminoglycosides did not result in significant changes in mortality compared to those who received β-lactam monotherapy. Further, no significant differences were found in 60-day relapse and 30-day readmission, suggesting that adjunctive use of aminoglycosides provides no clinical benefit in critically ill patients with GN-BSI.

The practice of utilizing combination therapy is a controversial practice with varying justifications over the years for a number of different pathogens. This review specifically aimed to elucidate any possible benefit of adjunctive aminoglycoside therapy in critically ill patients with GN-BSI compared to β-lactam monotherapy. While there have been other studies addressing the use of combination therapy, many included patients without confirmed infections, critically ill and non-critically ill patients, patients with non-bacteremic infections, or otherwise heterogeneous patient populations.

In the treatment of GN-BSI in critically ill patients, providers are faced with a dilemma of balancing the need for prompt effective therapy with the risk of adverse drug events and development of antimicrobial resistance due to antimicrobial overuse. Historically, aminoglycosides have been used as an adjunctive agent in these complex infections, primarily due to in vitro data and dogmatic practice. However, there are growing concerns with aminoglycoside use due to poor outcomes when used as a primary or monotherapy agent [27,28,29,30]. Many factors likely contribute to these poor outcomes, including pharmacokinetic/pharmacodynamic limitations such as poor pulmonary penetration [31,32,33] and increased volume of distribution in sepsis [34,35], high rates of adverse drug events, and reduced susceptibility breakpoints [36], which may limit the likelihood of covering a resistant pathogen, even with combination therapy.

Randomized controlled trials dating back to the 1980s have evaluated β-lactams compared to β-lactams plus adjunctive aminoglycosides for severe GN infections, demonstrating a lack of enhanced efficacy [11,12,13,19,24,25,26,30]. A myriad of agents have been involved in these studies, many of which are no longer used in clinical practice or no longer used in this clinical capacity. This has been further supported through meta-analyses and systematic reviews demonstrating no benefit of adjunctive aminoglycosides [9,10,16,18,21,37,38,39]. Detractors of these studies may point to the agents used in these studies, patient risk factors, and growing rates of antimicrobial resistance as justification to use adjunctive aminoglycoside therapy.

The data are conflicting on the role of adjunctive aminoglycosides in certain high-risk populations. For example, Albasanz-Puig et al. found a decrease in 7-day mortality when an aminoglycoside was given in addition to β-lactam therapy compared to β-lactam monotherapy in hematological neutropenic patients with GN-BSI. However, the authors list some potential limitations in this retrospective cohort study, such as variation across institutions, which may not have been accounted for, in addition to the high utilization of piperacillin–tazobactam in the monotherapy arm, with concerns about the ability of susceptibility testing methods to accurately identify resistant pathogens [40].

Interestingly, a meta-analysis by Kumar et al. evaluating adjunctive aminoglycoside therapy in patients with sepsis and septic shock found that combination therapy carried a mortality benefit in patients with a projected failure rate of >25%, whereas combination therapy was associated with an increased risk of mortality when the projected failure rate was ≤15%. Overall, they found no benefit of combination therapy in the general population [41]. These findings were corroborated by Gutiérrez-Gutiérrez et al., who found that while no difference in mortality was detected in the overall cohort, there was a mortality benefit with combination therapy compared with monotherapy in patients with a high mortality score on presentation [42].

Another meta-analysis by Sjövall et al. also found no difference in outcomes, including mortality, among adult patients admitted to the ICU with severe sepsis. As previously mentioned, the authors noted a high degree of heterogeneity in antimicrobial agents and reported that all included studies were at high risk of bias. Notably, they attempted to account for temporal changes in practice by performing a subgroup analysis of newer trials using agents common to current practice, but still no differences were detected [43].

Like the present study, Ong et al. detected no mortality difference or resolution of shock, but did note an increased risk of nephrotoxicity when using adjunctive aminoglycosides. However, their prospective observation study excluded patients with pulmonary sources and included non-bacteremic patients [44]. Our study differed in that we evaluated patients with confirmed GN-BSI to isolate only the patients who would benefit most from combination therapy.

Based on the prior literature, it is evident that the timely initiation of antibiotics, effective against causative organisms, is essential for reducing the risk of mortality in the critically ill with bloodstream infections [22,45,46,47,48,49,50,51,52,53,54,55]. However, the results of this study make it less evident that the addition of an aminoglycoside reduces this risk of mortality.

Similarly to previous studies evaluating adjunctive aminoglycoside therapy, the present study found no benefit of adjunctive aminoglycosides in critically ill patients with GN-BSI. In addition to demonstrating a lack of benefit, this study reinforces previous data by highlighting the adverse event profile of aminoglycosides, with 37% of combination therapy patients developing acute kidney injury compared to 26.5% in the monotherapy group. While not statistically significant in this study, the 10% absolute difference should be considered prior to initiating adjunctive aminoglycoside therapy. Strengths of this study included well-balanced groups, as evidenced by the propensity-matched analysis. Additionally, this was a robust patient population of a specific demographic at high risk for poor outcomes when empiric therapy is deemed to be inadequate. Evaluation of the dose and duration of aminoglycoside therapy also indicates appropriate dosing using an extended interval dosing approach and cessation of therapy after susceptibilities returned. This further supports the notion that similarities in efficacy and clinically significant differences in safety outcomes occurred in the setting of appropriate use of aminoglycosides in accordance with consensus guidelines recommendations and, therefore, confirms our hypothesis that adjunctive aminoglycosides did not improve outcomes in critically ill patients with GN-BSI.

This study is not without limitations, which are largely associated with the retrospective nature of the study. Due to the early mortality associated with sepsis and septic shock, the 72 h survival threshold may have led to immortal time bias; however, due to the frequently unclear clinical picture in early mortality, it would have been difficult to account for other factors that may have been associated with early mortality beyond the receipt—or lack thereof—of adjunctive aminoglycosides. Additionally, lack of documentation in the medical record may have led to incomplete data collection regarding medical history, surgical plans, and insight into clinical decision making. Additionally, while this study was adequately powered, the overall sample size is still relatively small, and the study was conducted at a single institution, both of which may lead to some limitations in generalizability.

## 4. Materials and Methods

### 4.1. Study Cohort and Setting

This was a retrospective, propensity-matched cohort study of patients 18 years or older with GN-BSI admitted to the ICU at a large academic medical center between 1 January 2017 and 28 February 2023. Data were collected and analyzed by infectious disease (ID)-trained physicians and pharmacists or trainees overseen by ID faculty. This study was reviewed and approved by the local Institutional Review Board (IRB) and determined to be exempt and of minimal risk to participants.

### 4.2. Enrollment Criteria

Patients ≥ 18 years of age with at least 1 positive blood culture containing Gram-negative rods (GNRs) and admitted to 1 of 5 intensive care units (medical, shock trauma, cardiovascular, neurological, or surgical) were eligible for inclusion. Exclusions were made based on the following criteria: presence of Gram-positive organism (s) on blood culture, death within 72 h of index culture, and pregnant patients.

### 4.3. Outcomes

The primary endpoint of this study was 15-day all-cause mortality. Secondary outcomes included 30-day all-cause mortality, ICU-free survival, 30- and 60-day relapse, 30-day readmission, development of new resistance, and adverse drug effects (ADEs). Development of resistance was defined as isolation of the same pathogen at any site with resistance to at least 1 new antibiotic compared to the index culture. ADEs in this study were limited to the development of acute kidney injury (AKI), as defined by an increase in serum creatinine by ≥0.3 mg/dL within 48 h, 1.5 times the baseline serum creatinine, or the new requirement for renal replacement therapy (RRT).

### 4.4. Statistical Analysis

The sample size was calculated for the primary outcome of 15-day all-cause mortality based on prior studies demonstrating incidence rates of up to 20–40% for GN-BSI in critically ill patient populations. Early mortality in critically ill patients with GN-BSI has been shown to be increased by ineffective empiric coverage, with prior studies showing a 17–23% range. To achieve 80% power using a two-sided α = 0.05, with a treatment difference of 20%, a cohort of 68 adjunctive aminoglycoside patients and 136 β-lactam monotherapy patients was required.

All statistical analyses were performed using R Statistical Software (v4.3.1, R Core Team 2023, Vienna, Austria) in R Studio. All statistical tests were performed on both the total and propensity-matched cohorts. The proportion of patients experiencing the primary outcome of 15-day mortality was assessed using the Chi-squared test. Secondary categorical outcomes including 30-day mortality, 60-day relapse, 30-day readmission, and safety outcomes were analyzed using the Chi-squared test. Continuous secondary outcomes were analyzed using the independent *t*-test or Mann–Whitney U test for normally and non-normally distributed data, respectively. Time-to-event analyses were performed using Kaplan–Meier curves and the log-rank test. Results were considered statistically significant at an alpha level of less than 5%.

A propensity-matched analysis was conducted to control for potential confounding variables. The propensity score was calculated via multivariable logistic regression based on independent predictors of aminoglycoside receipt or 15-day mortality. The following covariates were included in the regression model to determine propensity scores: age, sex, weight, baseline Pitt bacteremia score, vasopressor use, cancer, baseline Charlson comorbidity index (CCI), hemodialysis at baseline, presence of MDRO, and source of infection. Patients receiving aminoglycosides were matched to those not receiving aminoglycosides with a 1:2 ratio using the nearest-neighbor method and a caliper of 0.2 times the standard deviation of the logit of the propensity score. Patients without a match were excluded from the propensity-matched analysis. Standard mean difference (SMD) was calculated for each variable in the matched cohort, and an SMD < 0.1 was considered a good match.

## 5. Conclusions

This study did not show a benefit of using adjunctive aminoglycosides for treating GN-BSI in critically ill patients. While adverse effects were not significantly higher in those who received aminoglycosides, careful consideration should be given when using these agents adjunctively. The findings of this study still require confirmation through future prospective trials, but these data add to the growing body of literature suggesting that adjunctive aminoglycoside use does not improve clinical outcomes in critically ill patients.

## Figures and Tables

**Figure 1 antibiotics-14-00497-f001:**
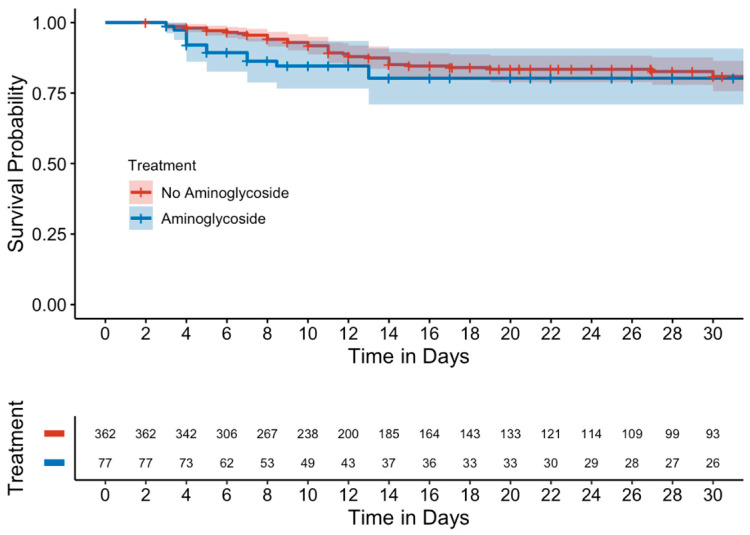
Kaplan–Meier curve of 15-day mortality (unmatched cohort).

**Figure 2 antibiotics-14-00497-f002:**
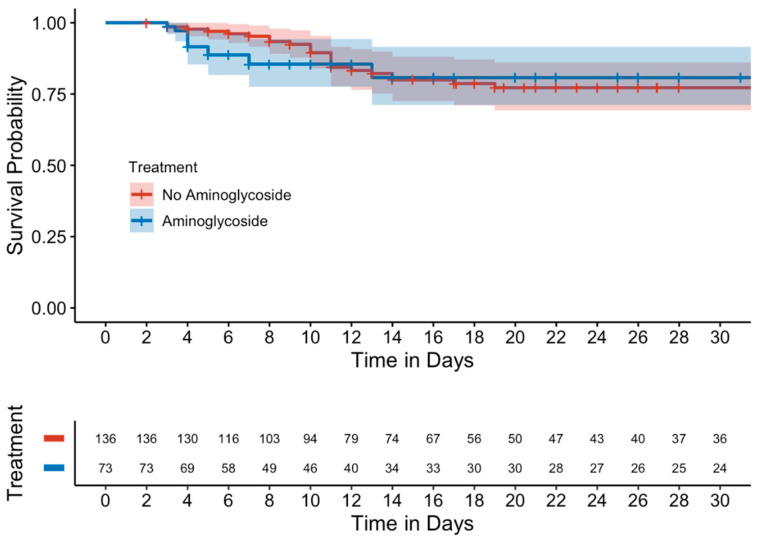
Kaplan–Meier curve of 15-day mortality (matched cohort).

**Figure 3 antibiotics-14-00497-f003:**
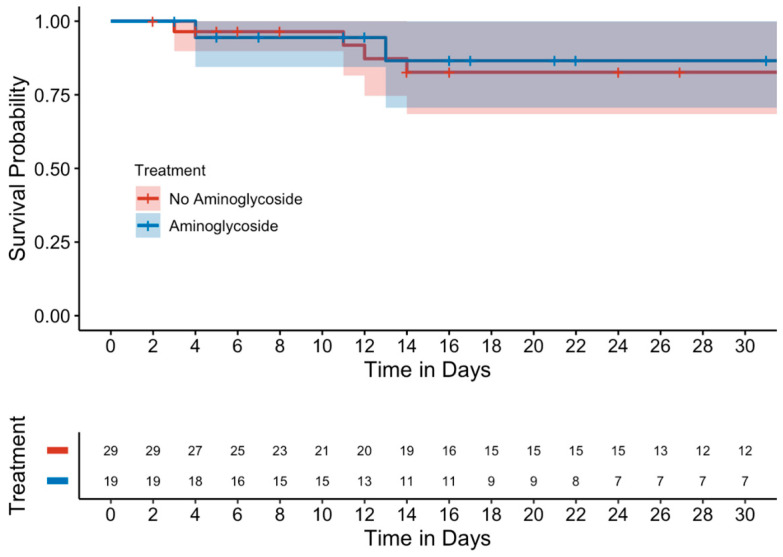
Kaplan–Meier curve of 15-day mortality in the MDRO subgroup (matched cohort).

**Table 1 antibiotics-14-00497-t001:** Demographics.

Variable	Unmatched Cohort			Matched Cohort		
	Monotherapy (362)	Aminoglycoside (77)	*p*-Value	Monotherapy (136)	Aminoglycoside (73)	SMD
Age (years), mean (SD)	60.46 (15.2)	60.75 (15.0)	0.880	61.0 (13.6)	60.8 (15.1)	0.014
Weight (kg), mean (SD)	83.4 (26.6)	83.2 (22.9)	0.954	83.0 (26.9)	82.0 (22.4)	0.037
Male, *n* (%)	201 (55.5)	41 (53.2)	0.811	69 (50.7)	39 (53.4)	0.054
CCI, mean (SD)	4.9 (3.09)	5.96 (3.37)	0.011	5.7 (3.2)	5.9 (3.4)	0.062
Liver disease, *n* (%)	42 (11.6)	14 (18.2)	0.167	20 (14.7)	13 (17.8)	0.084
Cancer, *n* (%)	75 (20.7)	27 (35.1)	0.011	42 (30.9)	24 (32.9)	0.043
Pitt, mean (SD)	3.6 (3.0)	4.66 (3.08)	0.004	4.3 (3.42)	4.6 (3.1)	0.038
Vasopressor use, *n* (%)	223 (61.6)	63 (81.8)	0.001	111 (81.6)	59 (80.8)	0.020
Pathogen information						
*Escherichia coli*, *n* (%)	142 (39.2)	27 (35.1)	0.581	48 (35.3)	24 (32.9)	0.051
*Klebsiella pneumoniae*, *n* (%)	74 (20.4)	19 (24.7)	0.502	30 (22.1)	18 (24.7)	0.061
*Pseudomonas aeruginosa*, *n* (%)	25 (6.9)	15 (19.5)	0.001	13 (9.6)	15 (20.5)	0.311
*Proteus mirabilis*, *n* (%)	34 (9.4)	3 (3.9)	0.177	13 (9.6)	3 (4.1)	0.217
*Serratia marcescens*, *n* (%)	25 (6.9)	6 (7.8)	0.976	11 (8.1)	6 (8.2)	0.005
*Enterobacter cloacae*, *n* (%)	20 (5.5)	3 (3.9)	0.764	7 (5.1)	3 (4.1)	0.049
*Acinetobacter* spp., *n* (%)	9 (2.5)	2 (2.6)	1.000	0 (0.0)	2 (2.7)	0.237
Other Enterobacterales, *n* (%)	32 (8.8)	2 (2.6)	0.104	14 (10.3)	2 (2.7)	0.310
Other NLF GNR, *n* (%)	1 (0.3)	0 (0.0)	1.000	0 (0.0)	0 (0.0)	-
Polymicrobial, *n* (%)	5 (1.4)	1 (1.3)	1.000	4 (2.9)	1 (1.4)	0.108
MDR organism, *n* (%)	54 (14.9)	22 (28.6)	0.007	29 (21.3)	19 (26.0)	0.111
Source of infection						
Urine, *n* (%)	151 (41.7)	25 (32.5)	0.169	48 (35.3)	24 (32.9)	0.051
Respiratory, *n* (%)	50 (13.8)	15 (19.5)	0.273	27 (19.9)	15 (20.5)	0.017
Intra-abdominal, *n* (%)	62 (17.1)	18 (23.4)	0.260	19 (14.0)	16 (21.9)	0.208
Primary BSI, *n* (%)	54 (14.9)	7 (9.1)	0.246	26 (19.1)	7 (9.6)	0.274
Central venous catheter, *n* (%)	13 (3.6)	3 (3.9)	1.000	3 (2.2)	3 (4.1)	0.109
SSTI, *n* (%)	27 (7.5)	8 (10.4)	0.528	12 (8.8)	7 (9.6)	0.026
Endovascular, *n* (%)	4 (1.1)	1 (1.3)	1.000	1 (0.7)	1 (1.4)	0.062
Bone and Joint, *n* (%)	1 (0.3)	0 (0.0)	1.000	0 (0.0)	0 (0.0)	-
Active therapy within 24 h, *n* (%)	313 (86.5)	62 (80.5)	0.161	117 (86)	61 (82.4)	0.099
Source control (yes or N/A), *n* (%)	74 (49.7)	20 (69.0)	0.089	27 (43.5)	20 (71.4)	0.588
Time to source control, mean (SD)	5.2 (7.06)	3.3 (3.0)	0.226	4.9 (8.2)	3.3 (3.0)	0.263

CCI: Charlson comorbidity index; NLF GNR: non-lactose-fermenting Gram-negative rod; MDR: multidrug-resistant; BSI: bloodstream infection; SSTI: skin and soft tissue infection.

**Table 2 antibiotics-14-00497-t002:** Aminoglycoside utilization.

Variable	
Aminoglycoside used	
Tobramycin, *n* (%)	7 (9.1)
Gentamicin, *n* (%)	70 (90.9)
Weight-based dose, mean (SD)	4.2 (1.6)
Time from index to first dose, median (IQR)	18.1 (9.2–23.9)

**Table 3 antibiotics-14-00497-t003:** Outcomes.

Outcome	Unmatched Cohort			Matched Cohort		
	Monotherapy (362)	Aminoglycoside (77)	*p*-Value	Monotherapy (136)	Aminoglycoside (73)	*p*-Value
15-Day Mortality, *n* (%)	46 (12.7)	16 (20.8)	0.096	23 (16.9)	15 (20.5)	0.644
30-Day Mortality, *n* (%)	64 (17.7)	19 (24.7)	0.206	30 (22.1)	18 (24.7)	0.800
ICU-Free Survival Days	19 (11.6)	17 (12.1)	0.275	16 (12.1)	17 (12.2)	0.813
60-Day Relapse *, *n* (%)	5 (1.8)	5 (8.8)	0.017	3 (3.3)	4 (7.4)	0.465
30-Day Readmission *, *n* (%)	65 (22.2)	12 (21.1)	0.989	24 (23.3)	10 (18.5)	0.626
New Resistance *, *n* (%)	19 (6.5)	5 (8.8)	0.730	12 (11.5)	5 (9.3)	0.867
AKI, *n* (%)	93 (25.7)	28 (36.4)	0.078	36 (26.5)	27 (37.0)	0.155

ICU: intensive care unit; AKI: acute kidney injury, * denominator includes only patients who survived beyond 30 days, not the overall cohort, to prevent early mortality from diluting results.

## Data Availability

The raw data supporting the conclusions of this article will be made available by the authors on reasonable request.

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
