# Peer review of "Evaluation of Adjunctive Aminoglycoside Therapy Compared to β-Lactam Monotherapy in Critically Ill Patients with Gram-Negative Bloodstream Infections"

_antibiotics, 2025, doi:10.3390/antibiotics14050497_

Round 1
Reviewer 1 Report
Comments and Suggestions for Authors
The study "Evaluation of adjunctive aminoglycoside therapy in critically ill patients with Gram-negative bloodstream infections" by Eudy et al, aims to evaluate the safety and efficacy of adjunctive aminoglycosides compared to β-lactam monotherapy in critically ill patients with gram-negative bloodstream infections (GN-BSI).
The authors found no significant mortality benefit of beta-lactam-aminoglycoside combination therapy compared to beta-lactam monotherapy for treating GN BSI in critically ill patients (17% vs. 21%; P=0.644). This finding aligns with previous studies that have questioned the clinical benefit of aminoglycoside combination therapy.
Although the study provides valuable insights, some critical issues need to be addressed to improve its clinical relevance:
- The most important issue is that the authors do not specify which beta-lactams were used in monotherapy (e.g. penicillins, cephalosporins or carbapenems), and which were used in combination with an aminoglycoside Furthermore, the authors do not report whether the same beta-lactam used in monotherapy was compared with its use in combination with an aminoglycoside, or whether a beta-lactam was compared with a different beta-lactam in combination with an aminoglycoside.
From a clinical point of view, the study's finding that there is no significant benefit of using β-lactam-aminoglycoside combination therapy compared to β-lactam monotherapy in critically ill patients with GN-BSI may have a different relevance if the drugs used in combination therapy and monotherapy differ in potency (e.g. if a carbapenem was used in monotherapy).
Accordingly, Table 1 should include which antibiotics were used in monotherapy and in combination with an aminoglycoside, and whether there are differences in use between the two groups, and the results should be discussed.
- The study uses a retrospective, propensity-matched cohort design, which inherently carries limitations such as potential selection and immortal time biases, as too briefly stated by the authors. Excluding patients who died within 72 hours of the index culture may have led to an underestimation of the early mortality associated with a specific antibiotic therapy (beta-lactam monotherapy or combination therapy with aminoglycosides). The authors should explain why these patients were excluded from the analysis and report the antibiotic treatment they received.
- The main outcome of the study is 15-day mortality, but the authors do not specify the time frame. Is it 15 days from admission to the ICU, from the collection/positivity of the index culture, or from the initiation of antibiotic therapy?
Is the antibiotic therapy initiated concomitantly with the collection of the index culture or with its positivity? In any case, which is the time to initiation of antibiotic therapy from admission? Are there any differences in the time of initiation between the groups?
- The study reports a higher incidence of acute kidney injury in the aminoglycoside group, raising concerns about the safety of adjunctive aminoglycoside therapy. The increased risk of nephrotoxicity associated with aminoglycosides is well documented in the literature and aligns with the study's finding. Although this finding was not statistically significant (26.5% vs. 37%; P=0.155), likely due to the small sample size, it may still be clinically significant and warrant further discussion. The potential for nephrotoxicity necessitates careful consideration, particularly in critically ill patients who are already at high risk of renal impairment, and this must be taken into account when deciding on the most appropriate treatment.
Other specific comments below:
Introduction
- Line 52: MDROs: Not initialed before in the text. Please check and standardise in the text.
- Lines 60-63: It is not clear what the authors mean. Please rephrase.
- Lines 64-67: Please concise.
Results
- Line 76: Figure 1 in brackets does not match the sentence in the text referring to Table 1. Please correct Figure 1 with Table 1.
- Line 78: SMD. Not initialed before in the text.
- Lines 82-85: What do the authors mean with “Primary BSI course”?
The authors state “Notable exceptions between the β-lactam monotherapy and adjunctive aminoglycoside groups included incidence of MDR organisms (21% vs. 26%; SMD 0.111), intra-abdominal source (14% vs. 22%; SMD 0.274), primary BSI source (19% vs. 10%; SMD 0.274), and source control (44% vs. 71%; SMD 0.588)”; although non statistically significant, these differences should be discussed.
- Line 91: Please rename Table 3 as Table 2 as the results you refer to appear first in the text and vice versa.
- Line 100: Please refer to the renamed Table 3, which is never mentioned in the text.
- Line 102: I think the reference is to Figure 1 and 2, not Figure 2 and 3. Is this correct? Otherwise, Figure 1 is not referred to in the text (see the first point, Line 76).
- Line 105: Please refer to Figure 3.
Tables
- Please add a legend to Tables 1 and 2 to explain the acronyms used (CCI, NLF GNR, MDR, SSTI, AKI).
- Line 115: Please rename Table 2 to Table 3. Please reverse the order.
- Line 119: Please rename table 3 to Table 2. Please reverse the order.
Discussion
- Lines 146-153: The authors refer to studies of monotherapy versus combination therapy, but do not clearly report the conclusions of these studies.
- Lines 157-161: Not clear. What the authors mean with “Practice sites”?
- Lines 183-186: This sentence sounds inappropriate with respect to the study data, as “timely initiation” of antibiotic therapy is not applicable to the study population, which includes only patients who survived after 72 See also point 2.
- Lines 199-203: It is not clear what the authors mean. Please rephrase. See also point 2.
Material and Methods
- Line 220: The authors should specify what they mean by 'index culture'. Does it refer to the first positive culture? To its collection date or to the date of its positivity? See point 3.
Conclusions (text and abstract)
- Line 262: In the present form of the manuscript the authors should specify that “This study did not show a benefit of using adjunctive aminoglycosides for treating GN-BSI in critically ill patients, who survived the first 72 hours”. This should be reconsidered in light of the answer to point 2.
Author Response
- The most important issue is that the authors do not specify which beta-lactams were used in monotherapy (e.g. penicillins, cephalosporins or carbapenems), and which were used in combination with an aminoglycoside Furthermore, the authors do not report whether the same beta-lactam used in monotherapy was compared with its use in combination with an aminoglycoside, or whether a beta-lactam was compared with a different beta-lactam in combination with an aminoglycoside.
From a clinical point of view, the study's finding that there is no significant benefit of using β-lactam-aminoglycoside combination therapy compared to β-lactam monotherapy in critically ill patients with GN-BSI may have a different relevance if the drugs used in combination therapy and monotherapy differ in potency (e.g. if a carbapenem was used in monotherapy).
Accordingly, Table 1 should include which antibiotics were used in monotherapy and in combination with an aminoglycoside, and whether there are differences in use between the two groups, and the results should be discussed.
Response: Thank you for this comment. While we understand the reviewers request, this information is not able to be adequately displayed as the use of antimicrobials are dynamic and difficult to differentiate and present data on empiric and definitive antimicrobials. There is ample data that the spectrum of activity has no correlation to clinical efficacy when treating susceptible pathogens. For example, a carbapenem would be not more efficacious than penicillin G for the treatment of group A streptococcal infections. Therefore, at our institution, with a robust antimicrobial stewardship program, patients are de-escalated to the most narrow-spectrum efficacious agent possible. For this reason, many patients may have been transitioned from cefepime, piperacillin/tazobactam, or a carbapenem to a more narrow-spectrum agent such as ceftriaxone, cefazolin, or even oral beta-lactams depending on pathogen identification and susceptibilities. For this reason, it is not feasible for us to report a singular beta-lactam used as monotherapy or in combination as these were frequently modified based on new culture data. The only independent variable was the presence or absence of aminoglycoside adjunctive therapy.
2. The study uses a retrospective, propensity-matched cohort design, which inherently carries limitations such as potential selection and immortal time biases, as too briefly stated by the authors. Excluding patients who died within 72 hours of the index culture may have led to an underestimation of the early mortality associated with a specific antibiotic therapy (beta-lactam monotherapy or combination therapy with aminoglycosides). The authors should explain why these patients were excluded from the analysis and report the antibiotic treatment they received.
Response: Thank you for this comment. While we agree that this exclusion may have led to the underestimation of early mortality, a survival threshold is commonplace for studies such as this as not including a survival threshold would have introduced a similar risk of bias, but the bias may have been more severe in its impact on the likelihood of introducing a Type II error. The attributable hypothesis is that patients who experience an early mortality outcome are likely to present with a higher severity of illness due to delayed implementation care from symptom onset. As a result, these patients are more likely to receive more aggressive therapies despite an unavoidable negative outcome. We introduced the propensity analysis for this purpose to ensure baseline similarities between groups while acknowledging that bias, while unavoidable was mitigated to the best of our abilities.
The main outcome of the study is 15-day mortality, but the authors do not specify the time frame. Is it 15 days from admission to the ICU, from the collection/positivity of the index culture, or from the initiation of antibiotic therapy?
Is the antibiotic therapy initiated concomitantly with the collection of the index culture or with its positivity? In any case, which is the time to initiation of antibiotic therapy from admission? Are there any differences in the time of initiation between the groups?
Response: The 15-day mortality was determined from index positivity. We have updated Table 1 to include rates of active therapy within 24 hours of index positivity.
4. The study reports a higher incidence of acute kidney injury in the aminoglycoside group, raising concerns about the safety of adjunctive aminoglycoside therapy. The increased risk of nephrotoxicity associated with aminoglycosides is well documented in the literature and aligns with the study's finding. Although this finding was not statistically significant (26.5% vs. 37%; P=0.155), likely due to the small sample size, it may still be clinically significant and warrant further discussion. The potential for nephrotoxicity necessitates careful consideration, particularly in critically ill patients who are already at high risk of renal impairment, and this must be taken into account when deciding on the most appropriate treatment.
Response Thank you for this comment, this is an excellent point. We have added a few comments in the discussion (lines 189-193) addressing the adverse event profile considerations for adjunctive aminoglycoside therapy.
Line 52: MDROs: Not initialed before in the text. Please check and standardise in the text
Response: Thank you. Addressed
6. Lines 60-63: It is not clear what the authors mean. Please rephrase
Response: Thank you. Addressed
7. Lines 64-67: Please concise.
Response: Thank you, addressed.
8. Line 76: Figure 1 in brackets does not match the sentence in the text referring to Table 1. Please correct Figure 1 with Table 1
Response: Thank you, addressed.
9. Line 78: SMD. Not initialed before in the text.
Response: Thank you, addressed.
10. Lines 82-85: What do the authors mean with “Primary BSI course”?.
The authors state “Notable exceptions between the β-lactam monotherapy and adjunctive aminoglycoside groups included incidence of MDR organisms (21% vs. 26%; SMD 0.111), intra-abdominal source (14% vs. 22%; SMD 0.274), primary BSI source (19% vs. 10%; SMD 0.274), and source control (44% vs. 71%; SMD 0.588)”; although non statistically significant, these differences should be discussed.
Response: We intended primary BSI as the source of infection, ie no other identifiable source was identified. These differences are acknowledged, but given the lack of impact on outcomes, we felt further commentary on these detract from the overall manuscript as any commentary would be hypothetical rather than data driven.
11. Line 91: Please rename Table 3 as Table 2 as the results you refer to appear first in the text and vice versa
Response: Thank you, addressed
12. Line 100: Please refer to the renamed Table 3, which is never mentioned in the text.
Response: Thank you, addressed
13. Line 102: I think the reference is to Figure 1 and 2, not Figure 2 and 3. Is this correct? Otherwise, Figure 1 is not referred to in the text (see the first point, Line 76).
Response: Thank you, addressed
14. Line 105: Please refer to Figure 3.
Response: Thank you, addressed.
15. Please add a legend to Tables 1 and 2 to explain the acronyms used (CCI, NLF GNR, MDR, SSTI, AKI)
Response: Thank you, addressed
16. Line 115: Please rename Table 2 to Table 3. Please reverse the order.
Response: Thank you, addressed
17. Line 119: Please rename table 3 to Table 2. Please reverse the order.
Response: thank you, addressed
18. Lines 146-153: The authors refer to studies of monotherapy versus combination therapy, but do not clearly report the conclusions of these studies
Response: Thank you for this comment, we have modified to note that these studies did not demonstrate a benefit of adjunctive aminoglycosides
19. Lines 157-161: Not clear. What the authors mean with “Practice sites”?
Response: we have changed the wording to “institutions”
20. Lines 183-186: This sentence sounds inappropriate with respect to the study data, as “timely initiation” of antibiotic therapy is not applicable to the study population, which includes only patients who survived after 72 See also point 2.
Response: thank you for this comment. Respectfully, we would insist on leaving this commentary. The common theme in BSI literature, including those evaluating combination therapy, is that the predictor for mortality is prompt initiation of active therapy. 72 hour mortality exclusion does not diminish the importance of prompt therapy, it simply reduces the risk of outcomes bias by excluding patients who would have a mortality outcome regardless of other interventions such as antimicrobial therapy.
21. Lines 199-203: It is not clear what the authors mean. Please rephrase. See also point 2.
Response: Thank you, we have modified the wording to highlight that the results were not due to inappropriate aminoglycoside dosing. We trust this is satisfactory.
22. Line 220: The authors should specify what they mean by 'index culture'. Does it refer to the first positive culture? To its collection date or to the date of its positivity? See point 3.
Response: Thank you for this comment. Index culture is widely used in the literature to indicate first culture that would qualify the subject for study inclusion. Given it’s usage in medical literature, we would prefer to leave the reference to index culture.
23. Line 262: In the present form of the manuscript the authors should specify that “This study did not show a benefit of using adjunctive aminoglycosides for treating GN-BSI in critically ill patients, who survived the first 72 hours”. This should be reconsidered in light of the answer to point 2.
Response: thank you for this comment. We prefer to keep our conclusions as stated. Adding a caveat for surviving 72 hours would imply waiting 72 hours to start therapy. Those patients were excluded to prevent skewing of results that would have favored the monotherapy group. Their exclusion reinforces the findings and is not a contingency as mentioned here.
Reviewer 2 Report
Comments and Suggestions for Authors The paper is very interesting and well organized: with a large study cohorts, correct enrollment criteria, interesting results. In particular it is the use of two different antibiotics that do not give (unfortunately) synergistic or additive results.The most interesting result is the lack of additive and benefit effect between the two antibiotics, which theoretically should at least give a cumulative effect. AS Authors say in the Concusions "this data adds to the growing body of literature that adjiunctive aminoglycosides use does not inprove clinical outocomes
in critically ill patiens"The references cited in this manuscript are appropriate and relevant to this research. Es.: Martinez JA. et al. Influence of empiric therapy with a beta-lactam alone or combined with an aminoglycoside on prognosis of bacteriemia due to gram-negative microorganisms. Antimicrob Agents Chemother. 2010;54(9);3590-6. Doi: 10.1128/AAC.00115-10.
Author Response
The paper is very interesting and well organized: with a large study cohorts, correct enrollment criteria, interesting results. In particular it is the use of two different antibiotics that do not give (unfortunately) synergistic or additive results.
The most interesting result is the lack of additive and benefit effect between the two antibiotics, which theoretically should at least give a cumulative effect. AS Authors say in the Concusions "this data adds to the growing body of literature that adjiunctive aminoglycosides use does not inprove clinical outocomes
in critically ill patiens"The references cited in this manuscript are appropriate and relevant to this research. Es.: Martinez JA. et al. Influence of empiric therapy with a beta-lactam alone or combined with an aminoglycoside on prognosis of bacteriemia due to gram-negative microorganisms. Antimicrob Agents Chemother. 2010;54(9);3590-6. Doi: 10.1128/AAC.00115-10.
Response: thank you for the kind words and for taking the time to review our manuscript.
Reviewer 3 Report
Comments and Suggestions for Authors
This manuscript is acceptable after major revisions. Below are my comments:
1. The reason to avoid using fluoroquinolones is due to the increasing rates of resistance. Have the authors considered the resistance to aminoglycosides?
2. Since the results of this study make it less evident that the addition of an aminoglycoside reduces the risk of mortality, what is the recommended strategy to reduce this risk?
3. Although adjunctive aminoglycoside use does not improve clinical outcomes in critically ill patients, have the authors considered any other drugs for improvement?
4. The references are outdated. Please add more recent references to support this study.
Author Response
1.The reason to avoid using fluoroquinolones is due to the increasing rates of resistance. Have the authors considered the resistance to aminoglycosides?
Response: Thank you for this comment. Yes, increasing rates of resistance to aminoglycosides was one of the reasons we felt this was a pertinent research endeavor to pursue. As we note in our introduction “In part, this could stem from pharmacokinetic/pharmacodynamic [PK/PD] issues which have led to recent breakpoint changes from the Clinical Laboratory and Science Institutes [CLSI] resulting in a significant increase in resistance to aminoglycosides.” This wording was changed slightly from the original manuscript to emphasize this point.
2. Since the results of this study make it less evident that the addition of an aminoglycoside reduces the risk of mortality, what is the recommended strategy to reduce this risk?
Response: We have attempted to address this in our discussion with commentary on the importance of timely, appropriate antimicrobial administration. Aminoglycoside combination therapy was the only independent variable we studied; therefore we did not feel we had sufficient data to be prescriptive with other recommendations beyond what previous literature has demonstrated in terms of time to active therapy.
3. Although adjunctive aminoglycoside use does not improve clinical outcomes in critically ill patients, have the authors considered any other drugs for improvement?
Response: We agree that future considerations is important due to the high mortality disease state studied. However, we tried to be as targeted as possible with our study to reduce confounders. Comparative studies may be considered by our research group in future endeavors, but we lack the data presently to support any specific treatment modalities.
4. The references are outdated. Please add more recent references to support this study.
Response: While we acknowledge that many of the references are outdated, they have significantly impacted this dogmatic practice that has been proliferated through the years. We attempted to be very thorough in our literature review and included many manuscripts published within the last five years. The lack of more recent data in light of changes in clinical practice and medical knowledge was one of the important considerations we thought was impactful when we planned and conducted the present study.
Reviewer 4 Report
Comments and Suggestions for Authors
This study takes a clinically relevant look at whether adding aminoglycosides to standard β-lactam therapy improves outcomes for critically ill patients with gram-negative bloodstream infections (GN-BSI). Using a well-matched retrospective cohort design, the authors found no meaningful difference in mortality, relapse, ICU recovery, or kidney injury between patients who received combination therapy and those who received monotherapy. Overall, this study provides an interesting insight into clinical antibiotic management for GN-BSI. Please kindly find the following comments and suggestions for improvement:
- Title: please improve clarity by stating beta-lactam monotherapy vs. combined therapy with adjunctive aminoglycosides
- Introduction: Strengthen the introduction by more clearly explaining why aminoglycosides are still considered for adjunctive therapy in the ICU despite known toxicity and limited outcome benefits in prior studies.
- Citations: Several references are clustered together in large groups (e.g., lines 54–63). These should be broken up and assigned to specific claims for clarity and precision.
- Demographic data: Please discuss the "cancer" effects on findings that were statistically significant between monotherapy vs. combined therapy groups.
- Figures: Figure 1-3 can be grouped into one to minimize stacked figures in the manuscript body. Furthermore, Kaplan-Meier survival curves should include hazard ratios and p-values directly in the figures or captions for easy interpretation.
- Table 3: not informative. The info could have been incorporated in the manuscript. Why is the total of aminoglycosides treatment not 100%?
- Discussion: has there been previous in vitro analysis regarding the bacteriocidal effects of the investigated aminoglycosides on GN-BSI bacteria? In addition, indicate aminoglycoside susceptibility rates among the isolated pathogens in each group to help readers assess the relevance of empiric aminoglycoside coverage.
- Defining study limitations: acknowledge more explicitly in the discussion that this is a single-center study with institution-specific protocols, which may limit broader applicability of the findings.
- Conclusion: The study concludes that using aminoglycosides along with β-lactams did not improve outcomes in critically ill patients with GN-BSI. Based on this, do the authors suggest using only β-lactam antibiotics for treatment? Also, can the authors comment on the cost difference between using β-lactam alone versus in combination with aminoglycosides, especially for hospitals concerned about treatment costs and antibiotic use?
Thank you.
Author Response
1. Title: please improve clarity by stating beta-lactam monotherapy vs. combined therapy with adjunctive aminoglycosides
Response: Thank you, we have modified the title accordingly.
2. Introduction: Strengthen the introduction by more clearly explaining why aminoglycosides are still considered for adjunctive therapy in the ICU despite known toxicity and limited outcome benefits in prior studies.
Response: Thank you for this comment. Lines 51 – 70 are dedicated to the rationale for adjunctive therapy. We trust this explanation is sufficient to address this concern.
3. Citations: Several references are clustered together in large groups (e.g., lines 54–63). These should be broken up and assigned to specific claims for clarity and precision.
Response: Due to the volume of studies and similar outcomes, we elected to group studies together to avoid redundancy and superfluous commentary that we felt would detract from our findings. We trust this is acceptable as presented.
4. Demographic data: Please discuss the "cancer" effects on findings that were statistically significant between monotherapy vs. combined therapy groups.
Response: Based on the propensity-matched analysis, there was no difference in rate of cancer diagnosis between groups (30.9% vs 32.9% with SMD of 0.043 indicating no baseline difference). The p-value in the unmatched cohort was statistically significant, but that difference did not persist in the propensity-matched analysis which is what the outcomes were based on.
5. Figures: Figure 1-3 can be grouped into one to minimize stacked figures in the manuscript body. Furthermore, Kaplan-Meier survival curves should include hazard ratios and p-values directly in the figures or captions for easy interpretation.
Response: If acceptable to the editor, we would prefer to keep KM curves separate rather than grouped for ease of interpretation for readers and so they can be referenced in the body of the manuscript easier. Reporting HRs are not standard with KM curves unless a Cox proportional hazard model is constructed. Given our low number of outcomes and similarities on the KM curves, we would prefer to leave our results as presented.
6. Table 3: not informative. The info could have been incorporated in the manuscript. Why is the total of aminoglycosides treatment not 100%?
Response: We felt addressing in the body would leave to verbosity and confusion on the part of the readers. If acceptable to the editor, we feel the table format is easier to convey factors related to aminoglycoside utilization. I believe it does add up to 100% (N=77, gentamicin (70) + tobramycin (7) totals 77.
7. Discussion: has there been previous in vitro analysis regarding the bacteriocidal effects of the investigated aminoglycosides on GN-BSI bacteria? In addition, indicate aminoglycoside susceptibility rates among the isolated pathogens in each group to help readers assess the relevance of empiric aminoglycoside coverage.
Response: Bactericidal activity has minimal impact on outcomes as described in the literature so we did not focus on that PD aspect of aminoglycosides. We are not able to provide susceptibility data for aminoglycosides as CLSI changed susceptibility breakpoints during the time period evaluated. Therefore, what may have been susceptible early in the study would have been called resistance later in the study which would have confounded results. Therefore, we have elected not to report susceptibility interpretations. We trust this response is sufficient justification to address this comment.
8. Defining study limitations: acknowledge more explicitly in the discussion that this is a single-center study with institution-specific protocols, which may limit broader applicability of the findings.
Response: Thank you for this comment. We have included the following line: “Additionally, while we met power for this study, the overall sample size is still relatively small and the study was conducted at a single institution, both of which may lead to some limitations in generalizability” we trust this sufficiently addresses this concern.
9. Conclusion: The study concludes that using aminoglycosides along with β-lactams did not improve outcomes in critically ill patients with GN-BSI. Based on this, do the authors suggest using only β-lactam antibiotics for treatment? Also, can the authors comment on the cost difference between using β-lactam alone versus in combination with aminoglycosides, especially for hospitals concerned about treatment costs and antibiotic use?
Response: Thank you for this comment. Yes, this interpretation of results is correct. While an important consideration, cost was outside the scope of this manuscript. While aminoglycosides are generally not cost prohibitive, it would be hard to quantify and control indirect costs associated with use such as AKI, need for dialysis, etc. with a high degree of accuracy.
Round 2
Reviewer 3 Report
Comments and Suggestions for Authors
This manuscript is acceptable.
Reviewer 4 Report
Comments and Suggestions for Authors
The authors have addressed the previously-given comments and suggestions for improvement.